# A Simulation-Free Deep Learning Approach to Stochastic Optimal Control

## Abstract

We propose a simulation-free algorithm for the solution of generic problems in stochastic optimal control (SOC). Unlike existing methods, our approach does not require the solution of an adjoint problem, but rather leverages Girsanov theorem to directly calculate the gradient of the SOC objective on-policy. This allows us to speed up the optimization of control policies parameterized by neural networks since it completely avoids the expensive back-propagation step through stochastic differential equations (SDEs) used in the Neural SDE framework. In particular, it enables us to solve SOC problems in high dimension and on long time horizons. We demonstrate the efficiency of our approach in various domains of applications, including standard stochastic optimal control problems, sampling from unnormalized distributions via construction of a Schrödinger-Föllmer process, and fine-tuning of pre-trained diffusion models. In all cases our method is shown to outperform the existing methods in both the computing time and memory efficiency.

## 1 Introduction

Stochastic Optimal Control (SOC) problems (Mortensen, 1989; Fleming & Rishel, 2012) arise in a wide variety of applications in sciences and engineering (Pham, 2009; Fleming & Stein, 2004; Zhang & Chen, 2022; Holdijk et al., 2023; Hartmann et al., 2013; 2017). Their general aim is to add an adjustable drift (the control) in a reference stochastic differential equation (SDE) so that the solutions of the controlled SDE minimize some cost.

In low dimensions, SOC problems can be tackled by standard numerical methods for partial differential equations since the optimal control can be obtained from the solution of a Hamilton-Jacobi-Bellman equation. In high dimensions, these methods are not applicable, and there has been a lot of recent effort to solve SOC problems via deep learning (DL), see e.g. (Han & E, 2016; Han et al., 2018; Huré et al., 2020; Domingo-Enrich et al., 2023; Germain et al., 2021; Hu & Lauriere, 2024).

The most direct way to leverage DL in the context of SOC is to parameterize the control by a deep neural network, view the SOC objective as a loss for the network parameters, evaluate this loss empirically using solutions of the controlled SDE, and use stochastic gradient descent to evolve the parameters until convergence. This fits into the framework of Neural SDE (Tzen & Raginsky, 2019; Li et al., 2020). While straightforward in principle, this approach faces a practical difficulty: since the SOC objective involves expectation over the solutions of an SDE that depend on the control, calculating the gradient of this objective requires differentiating through these solutions. This makes the training costly both in terms of computing time and memory requirement, and has prevented scaling up deep learning calculations of SOC policies.

A way to avoid this difficulty is to use Girsanov theorem to compute the SOC objective via an expectation over a reference process with a control that is independent from the control being optimized. Unfortunately, this requires to introduce an exponential weighing factor under the expectation to remove the bias incurred, and the variance of this factor is typically large if the control in the reference process is far from the actual control. This again limits the scalability of the approach.

In this paper, we propose a different procedure based on the observation that the gradient of the SOC objective over parameters in the control can be expressed exactly in terms of an expectation over the controlled process (which is referred to as *on-policy evaluation*) *without differentiating through the solution of this process*. We refer to this approach as *simulation-free*. This result, first derived in Yang & Kushner (1991), is obtained using Girsanov theorem, but it gives an expression for the gradient

that does not involve any exponential weighing factor. We also show that it is possible to calculate this gradient via automatic differentiation of an alternative objective, provided that we detach some of its parameters via stopgrad.

Specifically, our work makes the following **main contributions**:

- We propose a simulation-free, on-policy algorithm for solving generic SOC problems by deep learning and scaling up these calculations to situations where methods based on the Neural SDE framework is too costly to be applied.

- We discuss how to apply the method to construct Föllmer processes between a point mass and an unnormalized target distribution. Our approach can be used to sample this target and compute its normalization constant.

- We also discuss how to apply the method to fine-tune generative models based on diffusions: that is, assuming that we have at our disposal an SDE that samples a given distribution, we show how to find, via solution of a SOC problem, a new SDE that samples the distribution obtained by tilting the first one by a given reward function.

- We illustrate these results with numerical experiments showing that our method is much less expensive in terms of computing time and memory requirement than vanilla methods requiring differentiation through the solution of the SDE (such as in the Neural SDE framework).

## 1.1 RELATED WORK

Besides the traditional applications of SOC, such as finance and economics (Pham, 2009; Fleming & Stein, 2004; Aghion & Howitt, 1992), and robotics (Theodorou et al., 2011; Pavlov et al., 2018), this framework has recently has also found a variety of new applications, for example, sampling of complex non-log-concave probability distributions or simulation rare events (Zhang & Chen, 2022; Holdijk et al., 2023; Hartmann et al., 2013; 2017; Ribera Borrell et al., 2024). Our approach is aimed at solving problems of this sort by deep learning.

Deep learning for SOC has attracted a growing interest in the recent years. Notably, Han & E (2016) have proposed a deep learning method for high-dimensional SOC problems by learning the feedback control function. This method then inspired other algorithms to solve backward stochastic differential equations and partial differential equations (Han et al., 2018). Further developments can be found e.g. in (Ji et al., 2020), which studied three different algorithms. Another line of research proposed to use dynamic programming instead of learning the optimal control by simulating whole trajectories: see (Huré et al., 2021) for applications to SOC problems, and (Huré et al., 2020) for an extension to solving PDEs. Another recent approach is stochastic optimal control matching (Domingo-Enrich et al., 2023), based on iterative diffusion optimization (IDO) (Nüsken & Richter, 2023). Compared to the approach we propose, these methods either avoid differentiating through the SDE solution by using a independent reference process (off-policy learning), but this typically leads to estimators with high variance; or they use on-policy objectives, but it then requires to differentiate through the solution of the SDE. Our approach is on-policy but avoids this costly differentiation step.

A variant of formula for the gradient of the SOC objective that we used first appeared in (Yang & Kushner, 1991) in the context of sensitivity analysis with applications to finance and economics (Pham, 2009; Fleming & Stein, 2004; Aghion & Howitt, 1992). This formula is also discussed in Gobet & Munos (2005), who generalized it to problems in which the volatility also depends on the control. It is also mentioned in several works interested in solving SOC problem with DL (Mohamed et al., 2020; Li et al., 2020; Lie, 2021; Domingo-Enrich et al., 2023; Ribera Borrell et al., 2024; Domingo-Enrich, 2024), but to the best of our knowledge it has not been exploited systematically to develop the type of algorithm we propose here.

SOC also has deep connections with Reinforcement Learning (RL), see e.g. Quer & Borrell (2024); Domingo-Enrich et al. (2024); Domingo-Enrich (2024) for a discussion. The algorithm we propose can be viewed as a continuous-time variant of the well-known REINFORCE method (Williams, 1992; 1988; Sutton et al., 1999).

One application of SOC is the construction of a Föllmer process (Föllmer, 1986), i.e. a Schrödinger bridge between a Dirac delta distribution and a target distribution with unknown normalization coefficient. The Path Integral Sampler (PIS) proposed in Zhang & Chen (2022) is a way to solve this

problem via DL. PIS offers an alternative to other deep learning methods that have been introduced to calculate Schrödinger-Föllmer process (Huang et al., 2021; Jiao et al., 2021; Vargas et al., 2023) . PIS performs the minimization the SOC objective on-policy, and requires to differentiate through the solution of the controlled process. In contrast, our simulation-free approach performs on-policy minimization of the same objective, but it avoids this differentiation step which, as we will show below, results in significant gains in terms of memory requirement and computing time.

Another context in which SOC problems have found applications is the fine-tuning of generative models (Fan et al., 2024; Clark et al., 2024; Uehara et al., 2024). The aim here is to start from a diffusion model trained to sample a given data distribution and adjust the drift in this model so that it samples the probability distribution obtained by tilting the first by a reward function. Assuming that this reward function is differentiable, a method termed Adjoint Matching was recently proposed in Domingo-Enrich et al. (2024) to perform this fine-tuning: this method uses a specific scheduling of the noise in the diffusion and uses an adjoint method to compute the gradient of a tailored-built SOC objective. This second step avoids explicit differentiation through the solution of the controlled SDE by solving instead an ODE for an adjoint variable backward in time. Our simulation-free approach is an alternative to Adjoint Matching which uses the original SOC objective and avoid solving the ODE for the adjoint. It also works for any noise schedule, but requires that the base distribution used in the generative model be a point mass distribution.

## 2 METHODS

### 2.1 PROBLEM SETUP

We consider the following stochastic optimal control (SOC) problem:

$$\min_{u \in \mathcal{U}} J(u), \tag{1}$$

where the objective function is

$$J(u) = \mathbb{E}_{X^u} \left[ \int_0^T \left( \tfrac{1}{2} |u_t(X_t^u)|^2 + f_t(X_t^u) \right) dt + g(X_T^u) \right], \tag{2}$$

with $(X_t^u)_{t \in [0,T]}$ solution to the stochastic differential equation (SDE)

$$dX_t^u = (b_t(X_t^u) + \sigma_t u_t(X_t^u)) \, dt + \sigma_t dW_t, \qquad X_0^u \sim \mu_0. \tag{3}$$

In these equations, $X_t^u \in \mathbb{R}^d$ is the system state, $\mathbb{E}_{X^u}$ denotes expectation over the law of $(X_t^u)_{t \in [0,T]}$, $u : [0,T] \times \mathbb{R}^d \to \mathbb{R}^d$ is a closed-loop Markovian control that belongs to some set $\mathcal{U}$ of admissible controls to be specified later, $f : [0,T] \times \mathbb{R}^d \to \mathbb{R}^d$ is the state cost, $g : \mathbb{R}^d \to \mathbb{R}^d$ is the terminal cost, $b : [0,T] \times \mathbb{R}^d \to \mathbb{R}^d$ is the base drift, $\sigma : [0,T] \to \mathbb{R}^d \times \mathbb{R}^d$ is the volatility matrix, which we assume invertible and independent of the state $X_t^u$, $(W_t)_{t \in [0,T]}$ is a Wiener process taking values in $\mathbb{R}^d$, and $\mu_0$ is some probability distribution for the initial state.

We are interested in solving (1) in situations where the set of admissible controls $\mathcal{U}$ is a rich parametric class, for example made of deep neural networks (DNN). We denote functions in the class by $u^\theta$, where $\theta \in \Theta$ collectively denotes the parameters to be adjusted, e.g. the weights if we use a DNN.

### 2.2 REFORMULATION WITH GIRSANOV THEOREM

The main difficulty we face when solving the SOC problem (1) is that the process $X_t^u$ depends on the control $u$ since this control enters the SDE (3). In particular, vanilla calculation of the gradient of the objective requires to differentiate $X_t^u$: for example, if we parameterize $u = u^\theta$, we need to back-propagate the gradient through the solution of the SDE (3), which can be very costly. Our aim is to avoid this step and design a simulation-free method to compute the gradient of (2).

To this end, our starting point wil be to use Girsanov theorem to rephrase the problem in a way that render the control $u$ explicit in the objective:

**Lemma 1.** *Given a reference control $v \in \mathcal{U}$, the objective (2) can be expressed as*

$$J(u) = \mathbb{E}_{X^v} \left( \left[ \int_0^T \left( \tfrac{1}{2} |u_t(X_t^v)|^2 + f_t(X_t^v) \right) dt + g(X_T^v) \right] M(u,v) \right), \tag{4}$$

where $(X_t^v)_{t \in [0,T]}$ solves the SDE (3) with $u$ replaced by $v$, $M(u,v)$ is the Girsanov factor

$$M(u,v) = \exp\left(-\int_0^T (v_t(X_t^v) - u_t(X_t^v)) \cdot dW_t - \frac{1}{2}\int_0^T |v_t(X_t^v) - u_t(X_t^v)|^2 \, dt\right), \quad (5)$$

and the expectation $\mathbb{E}_{X^v}$ in (4) is taken over the law of $(X_t^v)_{t \in [0,T]}$.

*Proof.* The result is a direct consequence of Girsanov change of measure formula between the law of the SDE (3) and of its variant with $u$ replaced by $v$. □

The main advantage of expression (4) for the objective $J(u)$ is that it is explicit in $u$ since $(X^v)_{t \in [0,T]}$ is independent of this control (which is referred to as an off-policy objective). As a result, the evaluation of the gradient of (4) with respect to $u$ no longer requires differentiation through the trajectory of the state process. Of course, if we evaluate empirically $J(u)$ and its gradient via (4) by replacing the expectation $\mathbb{E}_{X^v}$ by an empirical expectation over a finite set of independent solution of the SDE for $X_t^v$, the variance of the resulting estimator will depend crucially on the reference control $v$ that we use. This suggests to keep this reference control $v$ close to the actual control $u$. Next, we show that we can actually use (4) to evaluate the gradient of $J(u)$ using the actual process.

## 2.3 GRADIENT COMPUTATION

The method proposed in this paper is the following formula for the gradient of $J(u)$ when we parameterize $u = u^\theta$. A similar formula first appeared in Yang & Kushner (1991) and it is also given in Ribera Borrell et al. (2024):

**Proposition 1.** *Let $u^\theta$ with $\theta \in \Theta$ be a parametric realization of a control in $\mathcal{U}$ and denote $L(\theta) \equiv J(u^\theta)$ the objective (2) viewed as a function of $\theta$. Then*

$$\partial_\theta L(\theta) = \mathbb{E}_{X^\theta}\left[\int_0^T u_t^\theta(X_t^\theta) \cdot \partial_\theta u_t^\theta(X_t^\theta)dt\right]$$
$$+ \mathbb{E}_{X^\theta}\left[\left(\int_0^T \left(\tfrac{1}{2}|u_t^\theta(X_t^\theta)|^2 + f_t(X_t^\theta)\right)dt + g(X_T^\theta)\right)\int_0^T \partial_\theta u_t^\theta(X_t^\theta) \cdot dW_t\right] \quad (6)$$

*where $\partial_\theta u_t^\theta(X_t^\theta)$ denotes $\partial_\theta u_t^\theta(x)$ evaluated at $x = X_t^\theta$ and $(X_t^\theta)_{t \in [0,T]} \equiv (X_t^{u^\theta})_{t \in [0,T]}$ solves the SDE*

$$dX_t^\theta = \left(b_t(X_t^\theta) + \sigma_t u_t^\theta(X_t^\theta)\right)dt + \sigma_t dW_t, \qquad X_0^\theta \sim \mu_0, \quad (7)$$

*and the expectation $\mathbb{E}_{X^\theta}$ in (6) is taken over the law of $(X_t^\theta)_{t \in [0,T]}$.*

We stress that the gradient expression (6) avoids completely the need of differentiating through the trajectory of the state process $(X_t^\theta)_{t \in [0,T]}$. (6) requires that the volatility $\sigma_t$ be invertible (so that (3) is elliptic) and independent of the control: it can however be generalized to controlled-dependent volatilities via Malliavin calculus (Gobet & Munos, 2005).

*Proof.* Equation (6) follows from (4) by a direct calculation in which we first evaluate the gradient of the objective $L(\theta) = J(u^\theta)$ with the fixed reference control $v$, which gives:

$$\partial_\theta L(\theta) = \mathbb{E}_{X^v}\left(\left[\int_0^T u_t^\theta(X_t^v) \cdot \partial_\theta u_t^\theta(X_t^v)dt\right]M(u^\theta, v)\right)$$
$$+ \mathbb{E}_{X^v}\left[\left(\int_0^T \left(\tfrac{1}{2}|u_t^\theta(X_t^\theta)|^2 + f_t(X_t^\theta)\right)dt + g(X_T^\theta)\right)\right. \quad (8)$$
$$\left. \times \left(\int_0^T \partial_\theta u_t^\theta(X_t^v) \cdot dW_t + \int_0^T (v_t(X_t^v) - u^\theta(X_t^v)) \cdot \partial_\theta u_t^\theta(X_t^v) \cdot dW_t\right)M(u^\theta, v)\right].$$

Because (8) holds for any $v$, we can now evaluate it at $v = u^\theta$. Since $M(u^\theta, u^\theta) = 1$, this gives (6). □

---

**Algorithm 1** Simulation-Free On-Policy Training

---

1: **Initialize:** $n$ walkers, $K$ time steps, model parameters $\theta$ for $u^\theta$, gradient descent optimizer
2: **repeat**
3:     Set $\bar{\theta} = \text{stopgrad}(\theta)$
4:     Randomize time grid: $t_1, \ldots, t_K \sim \text{Uniform}(0, T)$
5:     Add $t_0 = 0$, $t_K = T$, and sort such that $0 = t_0 < t_1 < \cdots < t_{K-1} < t_K = T$
6:     Set $\Delta t_k = t_{k+1} - t_k$
7:     **for** each walker $i = 1, \ldots, n$ **do**
8:         Set $x_0^i \sim \mu_0$, $A_0^i = 0$, $\bar{A}_0^i = 0$, $\bar{B}_0^i = 0$, $C_0^i = 0$
9:         **for** $k = 0, \ldots, K-1$ **do**
10:            $\Delta W_k^i = \sqrt{\Delta t_k}\, \zeta_k^i$, where $\zeta_k^i \sim N(0, \text{Id})$
11:            $x_{t_{k+1}}^i = x_{t_k}^i + u_{t_k}^{\bar{\theta}}(x_{t_k}^i)\Delta t_k + \sigma_{t_k}\Delta W_k^i$
12:            $A_{t_{k+1}}^i = A_{t_k}^i + \frac{1}{2}|u_{t_k}^\theta(x_{t_k}^i)|^2\Delta t_k$
13:            $\bar{A}_{t_{k+1}}^i = \bar{A}_{t_k}^i + \frac{1}{2}|u_{t_k}^{\bar{\theta}}(x_{t_k}^i)|^2\Delta t_k$
14:            $\bar{B}_{t_{k+1}}^i = \bar{B}_{t_k}^i + f_{t_k}(x_{t_k}^i)\Delta t_k$
15:            $C_{t_{k+1}}^i = C_{t_k}^i + u_{t_k}^\theta(x_{t_k}^i) \cdot \Delta W_k^i$
16:         **end for**
17:     **end for**
18:     Compute $\hat{L}_n(\theta, \bar{\theta}) = n^{-1}\sum_{i=1}^n \left[ A_{t_K}^i + \left( \bar{A}_{t_K}^i + \bar{B}_{t_K}^i + g(x_{t_K}^i) \right) C_{t_K}^i \right]$.
19:     Compute $\partial_\theta \hat{L}(\theta, \bar{\theta})\big|_{\bar{\theta}=\theta}$ and take a step of gradient descent to update $\theta$.
20: **until** converged

---

### 2.4 ALTERNATIVE OBJECTIVE FOR IMPLEMENTATION

Equation (6) can be implemented to directly estimate the gradient of the objective $L(\theta) = J(u^\theta)$ by replacing the expectation $\mathbb{E}_{X^\theta}$ by an empirical expectation over an ensemble of independent realizations of the SDE 7. Alternatively, we can use automatic differentiation of an alternative objective (Ribera Borrell et al., 2024; Domingo-Enrich, 2024):

**Proposition 2.** *We have*

$$\partial_\theta L(\theta) = \partial_\theta \hat{L}(\theta, \bar{\theta})\big|_{\bar{\theta}=\theta}, \tag{9}$$

*where we defined*

$$
\begin{aligned}
\hat{L}(\theta, \bar{\theta}) = {}& \mathbb{E}_{X^{\bar{\theta}}} \left[ \int_0^T \tfrac{1}{2}|u_t^\theta(X_t^{\bar{\theta}})|^2 dt \right] \\
&+ \mathbb{E}_{X^{\bar{\theta}}} \left[ \left( \int_0^T \left( \tfrac{1}{2}|u_t^{\bar{\theta}}(X_t^{\bar{\theta}})|^2 + f_t(X_t^{\bar{\theta}}) \right) dt + g(X_T^{\bar{\theta}}) \right) \int_0^T u_t^\theta(X_t^{\bar{\theta}}) \cdot dW_t \right]
\end{aligned}
\tag{10}
$$

*in which* $(X_t^{\bar{\theta}})_{t\in[0,T]} \equiv (X_t^{u^{\bar{\theta}}})_{t\in[0,T]}$ *solves (7) with* $u^\theta$ *replaced by* $u^{\bar{\theta}}$ *and the expectation* $\mathbb{E}_{X^\theta}$ *in (6) is taken over the law of this process.*

The proof of this proposition is immediate by direct calculation so we omit it for the sake of brevity. Note that can express the objective (10) as

$$\hat{L}(\theta, \bar{\theta}) = \mathbb{E}_{\bar{X}} \left[ A_T^\theta + \left( \bar{A}_T + \bar{B}_T + g(\bar{X}_T) \right) \bar{C}_T^\theta \right] \tag{11}$$

by defining

$$
\begin{aligned}
d\bar{X}_t &= b_t(\bar{X}_t)dt + \sigma_t u_t^{\bar{\theta}}(\bar{X}_t)dt + \sigma_t dW_t, & \bar{X}_0 &\sim \mu_0 \\
dA_t^\theta &= \tfrac{1}{2}|u_t^\theta(\bar{X}_t)|^2 dt, & A_0^\theta &= 0 \\
d\bar{A}_t &= \tfrac{1}{2}|u_t^{\bar{\theta}}(\bar{X}_t)|^2 dt, & \bar{A}_0 &= 0 \\
d\bar{B}_t &= f_t(\bar{X}_t)dt, & \bar{B}_0 &= 0 \\
dC_t^\theta &= u_t^\theta(\bar{X}_t) \cdot dW_t, & C_0^\theta &= 0
\end{aligned}
\tag{12}
$$

where $\bar{X}_t \equiv X_t^{\bar{\theta}}$, we use the bar to denote quantities that depend only on $\bar{\theta}$ and are therefore detached from differentiation over $\theta$. The Wiener process $W_t$ used in the equations for $\bar{X}_t$ and $C_t^\theta$ is the same.

In practice, the calculation of $\partial_\theta \hat{L}(\theta, \bar{\theta})\big|_{\bar{\theta}=\theta}$ can be implemented using automatic differentiation by setting $\bar{\theta} = \text{stopgrad}(\theta)$. This avoids again differentiation of $\bar{X}_t \equiv X_t^{\bar{\theta}}$ despite working with an on-policy objective. In practice we can again replace the expectation $\mathbb{E}_{X^\theta}$ by an empirical expectation over an ensemble of independent realizations of the equations in (12). This leads to the training method summarized in Algorithm 1, which involves one forward pass to evaluate the loss and avoids the need of memorizing any trajectory.

## 2.5 APPLICATION TO SAMPLING VIA CONSTRUCTION OF A FÖLLMER PROCESS

By definition, the Föllmer process that samples a given target probability distribution $\mu$ is the process $(Y_t^u)_{t \in [0,1]}$ that uses the optimal control $u$ obtained by solving

$$\min_{u \in \mathcal{U}} \mathbb{E}_{X^u} \int_0^1 \tfrac{1}{2} |u_t(Y_t^u)|^2 dt \tag{13}$$
$$\text{where:} \quad dY_t^u = u_t(Y_t^u)dt + dW_t, \qquad Y_0^u \sim \delta_0, \quad Y_{t=1}^u \sim \mu.$$

This problem is a special case of the Schrödinger bridge problem (Léonard, 2014) when the base distribution is the Dirac delta distribution $\delta_0$, i.e. the point mass at $x = 0$.

The minimization problem in (15) is not a SOC problem of the type (1) because of the terminal condition in the SDE that enforce $Y_{t=1}^u \sim \mu$. Interestingly, it can however be shown (Léonard, 2014; Chen et al., 2014) that the Föllmer process can also be constructed by solving a SOC problem, an observation that was exploited by Zhang & Chen (2022). We recall this result as follows:

**Proposition 3.** *Assume that $\mu$ is absolutely continuous with respect of the Lebesgue measure and let its probability density function be $\rho(x) = Z^{-1} e^{-U(x)}$ where $U : \mathbb{R}^d \to \mathbb{R}$ is a known potential and $Z = \int_{\mathbb{R}^d} e^{-U(x)} dx < \infty$ is an unknown normalization factor. Consider the SOC problem using the objective*

$$J(u) = \mathbb{E}_{X^u} \left[ \int_0^1 \tfrac{1}{2} |u_t(X_t^u)|^2 dt - \tfrac{1}{2} |X_1^u|^2 + U(X_1^u) \right], \tag{14}$$

*where $(X_t^u)_{t \in [0,T]}$ solves the SDE*

$$dX_t^u = u_t(X_t^u)dt + dW_t, \qquad X_{t=0}^u \sim \delta_0. \tag{15}$$

*Then the process $(X_t^u)_{t \in [0,1]}$ obtained by using the optimal control minimizing (14) in the SDE (15) is the Föllmer process that satisfies $X_{t=1}^u \sim \mu$.*

We omit the proof of this proposition since it is a special case of Proposition 5 established below.

The solution to the SOC problem in Proposition 3 can be achieved with our approach, thereby offering a simulation-free implementation of the Path Integral Sampler (PIS) proposed by Zhang & Chen (2022) that avoids the extra cost of the differentiation through the solution of the SDE (15) needed in the original PIS. The computational advantage this offers will be illustrated via examples in Sec. 3.

Note that since we have dropped the terminal constraint in the SDE (15) and replaced it by a terminal cost in (14), we are no longer guaranteed that $X_{t=1}^u \sim \mu$ if we do not use the optimal $u$. We can however compute an unbiased expectation over the target $\mu$ with any control via reweighing using Girsanov theorem, a fact that was also exploited by Zhang & Chen (2022):

**Proposition 4.** *Consider the process $(X_t^u)_{t \in [0,T]}$ obtained by solving the SDE (15) with any (not necessary optimal) control $u$. Then, given any suitable test function $h : \mathbb{R}^d \to \mathbb{R}$, we have*

$$\int_{\mathbb{R}^d} h(x)\mu(dx) = Z^{-1} \mathbb{E}_{X^u} \left[ h(X_T^u) M(u) \right], \qquad Z = \int_{\mathbb{R}^d} e^{-U(x)} dx = \mathbb{E}_{X^u} \left[ M(u) \right], \tag{16}$$

*where we defined*

$$M(u) = (2\pi)^{d/2} \exp\left( -\int_0^1 \tfrac{1}{2} |u_t(X^u)|^2 dt - \int_0^1 u_t(X_t^u) \cdot dW_t + \tfrac{1}{2} |X_1^u|^2 - U(X_1^u) \right). \tag{17}$$

*In addition $M(u) = Z$ iff $u$ is the optimal control minimizing the SOC problem with objective (14).*

We also omit the proof of this proposition since it is a special case of Proposition 6 established below.

## 2.6 Application to fine-tuning

We can consider a variant of the problem in Sec. 2.5 aiming at fine-tuning a generative model. Suppose that we are given a process $(Y_t)_{t \in [0,T]}$

$$dY_t = b_t(Y_t)dt + \sigma_t dW_t, \qquad Y_0 \sim \delta_0, \tag{18}$$

where the drift $b$ has been tailored in such a way that $Y_{t=T} \sim \nu$ where $\nu$ is a given probability distribution. Learning such a $b$ can for instance be done using the framework of score-based diffusion models (Song et al., 2021) or stochastic interpolants (Albergo & Vanden-Eijnden, 2022; Albergo et al., 2023) tailored to building Föllmer processes (Chen et al., 2024). Assume that we would like to fine-tune this diffusion so that it samples instead the probability distribution

$$\mu(dx) = Z^{-1} e^{r(x)} \nu(dx), \qquad Z = \int_{\mathbb{R}^d} e^{r(x)} \nu(dx) < \infty, \tag{19}$$

obtained by tilting $\nu$ by the reward function $r : \mathbb{R}^d \to \mathbb{R}$ (assuming that this tilted measure is normalizable, i.e. $Z < \infty$). Such problems arise in the context of image generation where they have received a lot of attention lately. Our next result shows that it can be cast into a SOC problem.

**Proposition 5.** *Consider the SOC problem (1) with zero running cost, $f = 0$, and terminal cost set to minus the reward function, $g = -r$, in the objective (13). Assume also that the drift $b$ and the volatility $\sigma$ used in the SDE (3) are the same as those used in the SDE (18) that guarantee that $Y_{t=T} \sim \nu$. Then the solutions of the SDE (3) solved with the optimal $u$ minimizing this SOC problem and $X_{t=0}^u = 0$ are such that $X_{t=T}^u \sim \mu$.*

The proof of this proposition is given in Appendix A. Note that Proposition 3 follows from Proposition 5 as a special case if we set $b_t(x) = 0$, $\sigma_t = 1$, and $T = 1$, in which case $\nu = N(0, \text{Id})$, and we can set $r(x) = -U(x) + \frac{1}{2}|x|^2$ to target $\mu(dx) = Z^{-1} e^{-U(x)} dx$.

The SOC problem in Proposition 5 can again be solved in a simulation-free way using our approach, thereby offering a simple alternative to the Adjoint Matching method proposed in (Domingo-Enrich et al., 2024). Since in practice the learned control will be imperfect, we will need to reweigh the samples to get unbiased estimates of expectations over them. It can be done using this result:

**Proposition 6.** *Let $X_t^u$ solves SDE (3) from the initial condition $X_{t=0}^u = 0$ with an arbitrary (not necesary optimal) $u$ and with the drift $b$ and the volatility $\sigma$ that guarantee that the solutions the SDE (3) satisfy $Y_{t=T} \sim \nu$. Then given any suitable test function $h : \mathbb{R}^d \to \mathbb{R}$, we have*

$$\int_{\mathbb{R}^d} h(x)\mu(dx) = Z^{-1} \mathbb{E}_{X^u}\left[ h(X_T^u) M_r(u) \right], \quad Z = \int_{\mathbb{R}^d} e^{r(x)} \nu(dx) = \mathbb{E}_{X^u}\left[ M_r(u) \right] \tag{20}$$

*where we defined*

$$M_r(u) = \exp\left( -\int_0^T \tfrac{1}{2}|u_t(X^u)|^2 dt - \int_0^T u_t(X_t^u) \cdot dW_t + r(X_T^u) \right) \tag{21}$$

*In addition, $M_r(u) = Z$ iff $u$ is the optimal control specified in Proposition 5.*

The proof of this proposition is given in Appendix A. Proposition 4 follows from Proposition 6 as a special case if we set $b_t(x) = 0$, $\sigma_t = 1$, $T = 1$, and $r(x) = -U(x) + \frac{1}{2}|x|^2$.

## 3 Experiments

### 3.1 Quadratic Ornstein-Uhlenbeck Example

We also consider a more complicated case where the SOC objective includes a quadratic running cost with $f(x) = x^T P x$, $g(x) = x^T Q x$, $b_t(x) = Ax$, $\sigma_t = \sigma_0$, where $P, Q, A \in \mathbb{R}^d \times \mathbb{R}^d$. This type of SOC problems are often referred to as linear quadratic regulator (LQR) and they have closed-form analytical solution (Van Handel, 2007):

$$u_t^*(x) = -2\sigma_0^\top F_t x, \tag{22}$$

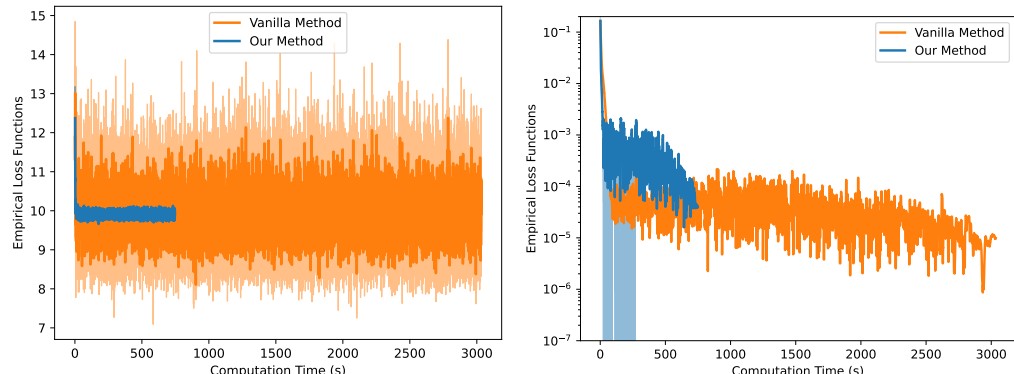

Figure 1: **Quadratic Ornstein-Uhlenbeck Example (easy setup, no warm-start)**: $L_2^2$ error (left panel) and training loss (right panel) for our method and the vanilla method. We plot the $95\%$ confidence intervals as shaded areas.

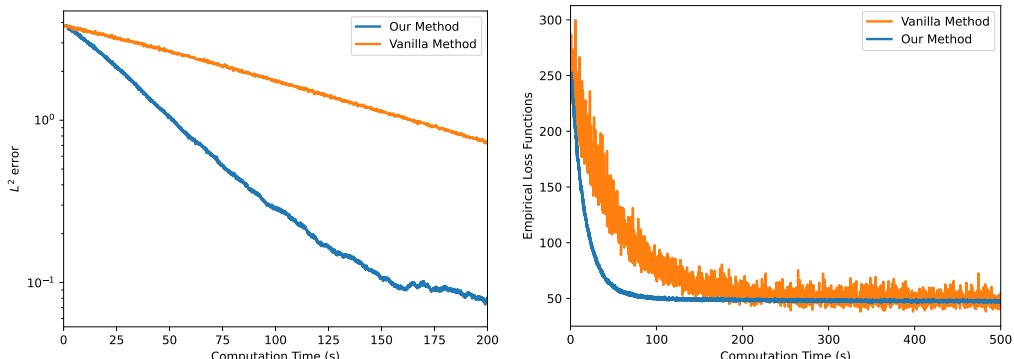

Figure 2: **Quadratic Ornstein-Uhlenbeck Example (hard setup, no warm-start)**: $L_2^2$ error (left panel) and training loss (right panel) for our method and the vanilla method.

where $F_t$ solves the Riccati equation

$$\frac{dF_t}{dt} + A^\top F_t + F_t A - 2|\sigma_0^\top F_t|^2 + P = 0 \tag{23}$$

with the final condition $F_T = Q$. We consider this example in the two setups investigated by Domingo-Enrich et al. (2023):

- *Easy:* $d = 20$, $A = 0.2I$, $P = 0.2I$, $Q = 0.1I$, $\sigma_0 = I$, $\lambda = 1$, $T = 1$, $X_0 \sim N(0, \frac{1}{2}\text{Id})$.
- *Hard:* $d = 20$, $A = I$, $P = I$, $Q = 0.5I$, $\sigma_0 = I$, $\lambda = 1$, $T = 1$, $X_0 \sim N(0, \frac{1}{2}\text{I})$.

However, in contrast to what was done in Domingo-Enrich et al. (2023) in the hard case, we use no warm-start. We use the same neural network parameterization and initialization for $u_t(x)$ as in Sec. B.1. We quantify the gain in efficiency in terms of the memory cost and computational cost of

| Model | Memory Cost (GB) | Run Time for Back-Prop (s) |
|---|---|---|
| Our Method | $\mathbf{0.102}_{\pm 0.001}$ | $\mathbf{0.002}_{\pm 0.003}$ |
| Vanilla Method | $0.160_{\pm 0.001}$ | $0.112_{\pm 0.0006}$ |

Table 1: **Quadratic Ornstein-Uhlenbeck Example (hard setup, no warm-start)**: Comparison between our model and the vanilla method in terms of the GPU memory usage and runtime for one back-propagation pass. Here, we use a batch size of $512$ and $256$ time steps.

| Model | Bias | Memory Cost (GB) | Run Time for Back-Prop (s) |
|---|---|---|---|
| Our Method ($\sigma_0 = 1.0$) | $-0.005$ $_{\pm 0.000}$ | **16.95** | **0.0032** $_{\pm 0.000}$ |
| ($\sigma_0 = 3.0$) | $0.032$ $_{\pm 0.005}$ | **16.95** | **0.0032** $_{\pm 0.000}$ |
| Path Integral Sampler ($\sigma_0 = 1.0$) | $-0.006$ $_{\pm 0.000}$ | $35.20$ | $1.014$ $_{\pm 0.0024}$ |
| ($\sigma_0 = 3.0$) | **0.029** $_{\pm 0.007}$ | $35.20$ | $1.014$ $_{\pm 0.0024}$ |

Table 2: **Funnel distribution example:** Comparison between our model and the PIS/Neural SDEs in terms of memory cost and runtime for $\sigma = 1.0, 3.0$. Here we use a batch size of $10^4$ and 400 time steps. For different experimental setups $\sigma_0 = 1.0$ and $\sigma_0 = 3.0$, the computational and memory cost are the same for the same method.

our method over the vanilla method in Table 1. In Figures 1 and 2 we compare both methods in terms of $L_2$ accuracy and training time for both the easy and the hard setups. As can be see, with the same time of compute, our method achieves a better accuracy measured by the $L_2$ error than the vanilla method. Training with our method is also more stable than with vanilla method, as the fluctuations in the empirical loss functions of our method are much smaller.

### 3.2 Sampling from an unnormalized distribution

Next, we consider an example of SOC problem relevant to the setup of Sec. 2.5. Specifically we sample Neal's funnel distribution in $d = 10$ dimension, for which $x_0 \sim N(0, \sigma_0)$ and $x_{1:9}|x_0 \sim N(0, e^{x_0} \mathrm{Id})$. This distribution was also used by Zhang & Chen (2022) to test their Path Integral Sampler (PIS): it becomes exponentially challenging to sample as $\sigma_0$ increases, as the spread $x_{1:9}$ is exponentially small for negative values of $x_0$ and exponentially large for positive values of $x_0$ (hence the funnel shape of the distribution), and the spread of $x_0$ around 0 increases with $\sigma_0$. We study this example with $\sigma_0 = 1$ and $\sigma_0 = 3$.

We use the same neural network parameterization as Zhang & Chen (2022), where the control is parameterized as follows

$$u_t^\theta(x) = \mathrm{NN}_1^\theta(t, x) + \mathrm{NN}_2^\theta(t) \times \nabla \log \rho(x). \tag{24}$$

Here $\rho(x)$ is the probability density function of the funnel distribution and $\nabla \log \rho(x)$ is its score. Note that the score function is the exact control we need if the time horizon $T \to \infty$. As what is reported in Zhang & Chen (2022), for one pass $\mathrm{NN}_1^\theta$ and $\mathrm{NN}_2^\theta$, we lift the scalar $t$ to 128 dimensions with Fourier positional encoding, followed by two fully connected layers with 64 hidden units to extract 64-dimensional signal features. We use an independent two-layer MLP to extract 64-dimensional features for $x$. Then, we concatenate the features of the two MLPs for $x, t$ and use it as the input to a three-layer neural network to obtain the output. To make the problem challenging, we initialize at zero the weights of the last linear layers of $\mathrm{NN}_1^\theta$ and $\mathrm{NN}_2^\theta$, so that $u_t^\theta(x) = 0$ at initialization. We use the Adam optimizer (Kingma, 2014) with a learning rate of $5 \cdot 10^{-3}$ for all the experiments on the funnel distribution example.

To measure the quality of the learned samplers, we estimate the log normalization constants $\log Z$ of the funnel distribution with $\sigma_0 = 1.0$ and $\sigma_0 = 3.0$ with the unbiased estimator given in (16). We report the results as well as the computational costs in Table 2: our method reaches the same level of accuracy as the vanilla PIS but with much smaller memory cost and computing time, which is a big advantage for problems in high dimensions. The generated samples themselves are shown in Figure 3.

## 4 Conclusion

We have introduced a simulation-free on-policy approach to SOC problems: we simulate trajectories using an actual control but detach this control for the computational graph when computing the gradient of the objective. This leads to a new way of training a deep neural network to learn the feedback control, which is more efficient and scalable than the vanilla method commonly used thus far. We have illustrated the efficacy of our method on several examples from the SOC literature and the sampling literature, with applications to sampling via Föllmer processes and fine-tuning of diffusion-based generative models .

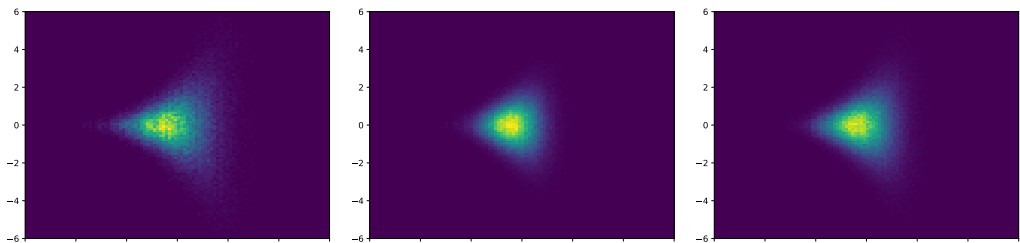

Figure 3: **Funnel distribution example** ($\sigma = 1$): The samples from the funnel distribution (left panel), Path Integral Samplers (middle panel) and our method (right panel). To plot the samples of the ten-dimensional funnel distribution in 2D, we use the independence of its coordinates $\{x_1, \cdots, x_9\}$, squeeze these nine dimensions into one coordinate, and keep the first dimension $x_0$.

**Limitations.** While our approach is simpler and more efficient than the vanilla method to minimize the SOC objective, it eventually performs the same minimization of the SOC objective. As a result, it is prone to the same issues: slow convergence of the training in absence of good warm-start and potential lack of convergence to the non-convexity of the SOC objective. Nevertheless the experiments reported here clearly show the potential of our approach for large-scale experiments on real-world problems, which we leave for future work.

## REPRODUCIBILITY STATEMENT

All experiments done in this work rely on simply feed-forward neural networks, and can be done locally on a single GPU. Details for the network sizes are given in each experimental subsection.

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

## A    PROOFS OF PROPOSITIONS 5 AND 6

*Proof of Proposition 5.* It is well-known (Léonard, 2014; Chen et al., 2014) that the SOC problem specified in the proposition can be cast into solving the pair of partial differential equations

$$\partial_t \mu_t = -\nabla \cdot (b_t \mu_t) - \nabla \cdot (D_t \nabla \phi_t \mu_t) + \tfrac{1}{2}\nabla \cdot (D_t \nabla \mu_t), \qquad \mu_0 = \delta_0 \qquad (25)$$

$$\partial_t \phi_t = -b_t \cdot \nabla \phi_t - \tfrac{1}{2}\nabla \phi_t \cdot D_t \nabla \phi_t - \tfrac{1}{2}\nabla \cdot (D_t \nabla \phi_t), \qquad \phi_T = r \qquad (26)$$

where $D_t = D_t^T = \sigma_t \sigma_t^T$, $\mu_t$ is the distribution of $X_t^u$, and the potential $\phi_t$ gives the optimal control via $u_t = \sigma_t^T \nabla \phi_t$. We also know that the distribution $\nu_t$ of $Y_t$ solves the Fokker-Planck equation

$$\partial_t \nu_t = -\nabla \cdot (b_t \nu_t) + \tfrac{1}{2}\nabla \cdot (D_t \nabla \nu_t), \qquad \nu_0 = \delta_0 \qquad (27)$$

We will prove that $X_T^u \sim \mu$ by establishing that $\mu_t(dx) = Z^{-1}e^{\phi_t(x)}\nu_t(dx)$ since this will imply that $\mu_T = \mu$ since $\nu_T = \nu$ by definition and $\phi_T = r$ by construction, so that $\mu_T = Z^{-1}e^{\phi_T}\nu_T = Z^{-1}e^r\nu = \mu$. If $\hat{\mu}_t(dx) = Z^{-1}e^{\phi_t(x)}\nu_t(dx)$, then we have

$$\partial_t \hat{\mu}_t = Z^{-1}\left(\partial_t \phi_t e^{\phi_t}\nu_t + e^{\phi_t}\partial_t \nu_t\right),$$

$$-\nabla \cdot (b_t \hat{\mu}_t) = Z^{-1}\left(-e^{\phi_t}\nabla \cdot (b_t \nu_t) - b_t \cdot \nabla \phi_t e^{\phi_t}\nu_t\right),$$

$$-\nabla \cdot (D_t \nabla \phi_t \hat{\mu}_t) = Z^{-1}\left(-e^{\phi_t}\nabla \cdot (D_t \nabla \phi_t \nu_t) - e^{\phi_t}\nabla \phi_t \cdot D_t \nabla \phi_t \nu_t\right)$$

$$= Z^{-1}\left(-e^{\phi_t}\nabla \cdot (D_t \nabla \phi_t)\nu_t - e^{\phi_t}\nabla \phi_t \cdot D_t \nabla \nu_t - e^{\phi_t}\nabla \phi_t \cdot D_t \nabla \phi_t \nu_t,\right) \qquad (28)$$

$$\tfrac{1}{2}\nabla \cdot (D_t \nabla \hat{\mu}_t) = Z^{-1}\left(\tfrac{1}{2}e^{\phi_t}\nabla \cdot (D_t \nabla \nu_t) + e^{\phi_t}\nabla \phi_t \cdot D_t \nabla \nu_t\right.$$

$$\left. + \tfrac{1}{2}e^{\phi_t}\nabla \phi_t \cdot D_t \nabla \phi_t \nu_t + \tfrac{1}{2}e^{\phi_t}\nabla \cdot (D_t \nabla \phi_t)\nu_t\right).$$

Inserting these expressions in (25) and using (27), several terms cancel we are left with

$$\partial_t \phi_t e^{\phi_t}\nu_t = -b_t \cdot \nabla \phi_t e^{\phi_t}\nu_t - \tfrac{1}{2}\nabla \phi_t \cdot D_t \nabla \phi_t e^{\phi_t}\nu_t - \tfrac{1}{2}e^{\phi_t}\nabla \cdot (D_t \nabla \phi_t)\nu_t. \qquad (29)$$

If we divide both sides of this equation by $e^{\phi_t}\nu_t$, we recover (26). This shows that $\hat{\mu}_t(dx) = Z^{-1}e^{\phi_t(x)}\nu_t(dx)$ is indeed a solution to (25). To show that it is the solution, it remains to establish that it satisfies the initial condition in (25). To this end, notice first that, since $\nu_0 = \delta_0$, we have

$$\hat{\mu}_0(dx) = Z^{-1}e^{\phi_0(x)}\nu_0(dx) = Z^{-1}e^{\phi_0(0)}\delta_0(dx) \qquad (30)$$

Second, since $\hat{\mu}_t$ satisfies (25), we must have $\int_{\mathbb{R}^d} \mu_t(dx) = 1$ for all $t \in [0, 1]$. As a result, we conclude that $e^{\phi_0(0)} = Z$, which means that $\hat{\mu}_0 = \delta_0 = \mu_0$. Since the solution pair $(\mu_t, \phi_t)$ to (25)-(26) is unique, we must have $\mu_t = \hat{\mu}_t = Z^{-1}e^{\phi_t}\nu_t$ and hence $\mu_T = Z^{-1}e^{\phi_T}\nu_T = Z^{-1}e^r\nu$. $\qquad \square$

Note that it is key that $\mu_0 = \nu_0 = \delta_0$ (more generally $\delta_{x_0}$ for some $x_0 \in \mathbb{R}^d$). If the base distribution used to generate initial data in the SDEs (3) and (18) are not atomic at $x = 0$, the statement of Proposition 5 does not hold anymore, because the second equality in (30) fails. That is, our framework only allows to fine-tune generative models that use a Dirac delta distribution as base distribution.

*Proof of Proposition 6.* By direct application of Girsanov theorem, we have

$$\mathbb{E}_{X^u}[h(X_T^u)M_r(u)] = \mathbb{E}_Y\left[h(Y_T)e^{r(Y_T)}\right] = \int_{\mathbb{R}^d} h(x)e^{r(x)}\nu(dx), \qquad (31)$$

| Model | Memory Cost (GB) | Run Time for Back-Prop (s) |
|---|---|---|
| Our Method | **1.962** $_{\pm 0.001}$ | **0.003** $_{\pm 0.000}$ |
| Vanilla Method | $2.590$ $_{\pm 0.001}$ | $0.177$ $_{\pm 0.0006}$ |

Table 3: **Linear Ornstein-Uhlenbeck Example**: Comparison between our method and the vanilla method in terms of the GPU memory usage and runtime for one back-propagation pass. Here, we use a batch size of 5k and 256 time steps.

where $Y_t$ solves (18) and we used $Y_T \sim \nu$ to get the second equality. Multiplying both sides of (31) by $Z^{-1}$ we deduce

$$Z^{-1}\mathbb{E}_{X^u}\left[h(X_T^u)M_r(u)\right] = Z^{-1}\int_{\mathbb{R}^d} h(x)e^{r(x)}\nu(dx) = \int_{\mathbb{R}^d} h(x)\mu(dx), \tag{32}$$

which gives the first equation in (20). Setting $h = 1$ in (31) we deduce that

$$\mathbb{E}_{X^u}\left[M_r(u)\right] = \int_{\mathbb{R}^d} e^{r(x)}\nu(dx) = Z \tag{33}$$

which gives the second equation in eq. (20). To establish that $M_r(u) = Z$ *iff* $u$ is the optimal control minimizing the SOC problem specified in Proposition 5, notice that $X_T^u \sim \mu$ *iff* $u$ is this optimal control. Assuming tat this is the case, the first equation in eq. (20) implies that

$$\mathbb{E}_{X^u}\left[h(X_T^u)\right] = Z^{-1}\mathbb{E}_{X^u}\left[h(X_T^u)M_r(u)\right] \tag{34}$$

for all suitable test function $h$. This can only hold if $M_r(u) = Z$. $\square$

# B  ADDITIONAL NUMERICAL RESULTS

## B.1  LINEAR ORNSTEIN-UHLENBECK EXAMPLE

We consider the SOC problem in (1) with $f = 0$, $g(x) = \gamma \cdot x$, $\sigma_t = \sigma = cst$, $\lambda = 1$ and $b_t(x) = Ax$, where $\gamma \in \mathbb{R}^d$ and $\sigma, A \in \mathbb{R}^d \times \mathbb{R}^d$. This example was proposed by Nüsken & Richter (2023) and its optimal control can be calculated analytically:

$$u_t^*(x) = -\sigma_0^\top \exp(A^\top(T-t))\gamma. \tag{35}$$

We take $d = 20$ and initialize the samples at $X_0 \sim \mathcal{N}(0, \frac{1}{2}\mathrm{Id})$. We use our simulation-free method to estimate the control $u_t(x)$ after parameterizing it by a fully connected MLP with 4 layers and 128 hidden dimensions in each layer. The parameters of this MLP network are initialized with the default random initializations of PyTorch. For comparison, we also minimize the SOC objective with the vanilla method that requires differentiation though the solution of the SDE, using the same exact setup. To make fair comparisons, we use the Adam optimizer (Kingma, 2014) with the same learning rate of $3 \cdot 10^{-4}$ and the same cosine annealing learning rate scheduler for both methods.

The numerical results are shown in Figure 4 and the squared $L_2$ error in each figure is computed by approximating the following quantity with Monte-Carlo estimations over 256 sample trajectories

$$E = \int_0^T \mathbb{E}_{X_t^*}\left[|u_t(X_t^*) - u_t^*(X_t^*)|^2\right] dt \tag{36}$$

where $X_t^*$ is the process generated with the optimal control $u^*$ and $X_0^* \sim \mathcal{N}(0, \frac{1}{2}\mathrm{Id})$.

As compared to the vanilla method, our method is both more computationally efficient and more accurate (see Table 3 for a detailed comparison in terms of memory cost and computational time).

## B.2  HIGH-DIMENSIONAL QUADRAIC OU EXPERIMENTS

We perform the quadratic OU control experiments done in Section 3.1 for dimension $D = 50$ and time horizon $T = 4.0$ with the hard setup and no warm-start. The results are plotted in Figure  where our method outperforms the vanilla method in terms of both accuracy and time efficiency.

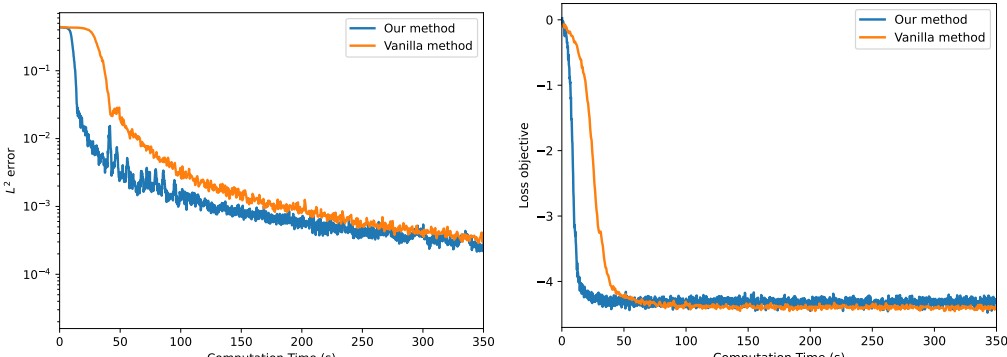

Figure 4: **Linear Ornstein-Uhlenbeck Example**: Our method outperforms the vanilla method in terms of accuracy measured by the squared $L_2$ error (left panel) and computational efficiency (right panel).

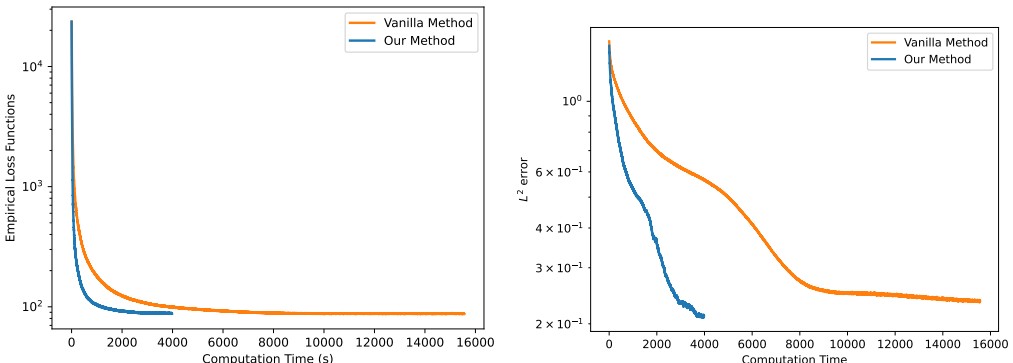

Figure 5: **Quadratic Ornstein-Uhlenbeck Example (hard setup, no warm-start)**: $L_2^2$ error (left panel) and training loss (right panel) for our method and the vanilla method. Here we extend the dimension to $D = 50$ and time horizon to $T = 4.0$.

