# OpenReview forum: "A Simulation-Free Deep Learning Approach to Stochastic Optimal Control"
_ICLR.cc/2025/Conference — Submitted to ICLR 2025_

### Official Review · Reviewer_SeZL · 2024-10-19

**Soundness:** 1
**Presentation:** 2
**Contribution:** 1
**Rating:** 3
**Confidence:** 4

**Summary:**

This paper proposes a method to solve stochastic optimal control problems by utilizing the Girsanov theorem to change the probability measure. It claims that this reformulation makes the policy objective explicit in terms of the control input, allowing existing optimization techniques to be applied directly.

**Strengths:**

* This paper attempts to solve an important problem.

* The presentation is clear and easy to follow.

* The authors have made efforts to present their algorithm in a rigorous manner

**Weaknesses:**

There are some critical errors in the paper:

* The core foundation of the proposed algorithm is the change of probability measure using the Girsanov theorem. However, there seems to be a conceptual confusion throughout the paper. When calculating the expectation in Equation (2), the probability measure is the one under which the random noise in the system is a Brownian motion. This measure is independent of the SDE of $X^u,$ it is determined by how the random noise in the stochastic system is measured. By changing the probability measure, it implies we are considering a different way of measuring the noise (e.g., applying a transformation to it), but this is clearly not the intention of the paper.

* I am uncertain about the correctness of Lemma 1, as the proof is omitted. Specifically, I do not see how the Girsanov factor $G(u, v)$ is derived. The probability measure is not changed from (2) to (4). Also, the notation $\mathbb{E}_{X^u}$ is unclear to me. All expectations in this paper should be the same expectation as in (2), since it is exactly how the trajectories are sampled from the environment, as shown in Algorithm 1.

* Proposition 1 appears to be incorrect. Assuming that Equation (4) is valid, Equation (8) is obtained by directly differentiating (4) with respect to the policy parameter $\theta$. At the end of the proof, the authors claim that (8) implies (6) because (8) holds for any fixed reference control $v$, hence we can just simply let $v = u$ which yields $M(u, v) = 1$. This reasoning is not true because "(8) holds for any fixed reference control $v$" DOES NOT mean that its value is independent of $v$.

* The experimental results are not convincing. First, all the problems presented in the experiments can be solved using either Monte Carlo methods or analytical solutions. Second, the proposed algorithm is only compared with the vanilla method. There are various existing algorithms that might be included for comparison, such as 'Solving high-dimensional partial differential equations using deep learning' by Han et al.

**Questions:**

Please see the weaknesses.

---

> ### Author Response · Authors · 2024-11-25
>
> We thank the reviewer for their comments. We do believe that the confusion is on the reviewer, however. Let us try to clarify our statements.
>
> **Weakness 1 (confusion about Girsanov theorem):**
> Lemma 1 is a standard application of Girsanov theorem. It does indeed involves a change of measure, but the result is well-known: in a nutshell, Girsanov theorem allows one to express expectations over the solution of one SDE as expectations over another SDE with a modified drift, provided that the Girsanov factor is added to remove the bias exactly, precisely as stated in our Lemma 1. (The reviewer may also want to take a quick look at the wikipedia page: https://en.wikipedia.org/wiki/Girsanov_theorem )
>
> To avoid confusion we will relabel this lemma as *Girsanov theorem*.
>
> **Weakness 2 (proof of Lemma 1):**
> We don't think it makes sense for us to add a proof of Girsanov theorem since it can be found in any standard textbook, such as e.g. Øksendal book *Stochastic Differential Equations: An Introduction with Applications* (Springer, 2003). We will add a reference to this book.
>
> **Weakness 3 (Proposition 1):**
> Here too, the reviewer seems confused for no reason. Proposition 1 is a direct consequence of Girsanov theorem, as our proof shows. We also note that, as pointed out by reviewer **2UrK** and mentioned by us in text, this proposition is not really new: it is a variant of a result first derived in Yang & Kushner (1991). Our paper mainly generalizes this result to  control parameterized by deep neural networks and show how to apply it in practice in this context.
>
> **Weakness 4 (Experimental results):**
> Most of the examples we used have analytical solutions indeed: they are however standard test cases for SOC numerical methods, and are used precisely because they allow one to benchmark the numerical solution. We also stress that these examples are by no means trivial to solve numerically, as they are high-dimensional: in fact, standard solution methods either fail or scale badly as the dimension of the problem increases.
>
>
> Regarding the baseline comparison with other methods, it is natural for us to consider mainly the vanilla method (i.e. the one requiring backpropagation trough the solution of the SDE), since it is the method of choice. Several other methods have appeared recently, in particular approaches that require solving an adjoint equation, but these schemes are much more complicated to implement than the one we propose, so we decided to leave a detailed comparison with these to future work.
>
> Finally, let us stress that the approach of Han et al. mentioned by the reviewer is a scheme to solve a PDE by rewriting its solution as the solution to a backward stochastic differential equation (BSDE), rewriting this BSDE as a control problem, and then applying the vanilla method to this control problem (with backpropagation through the SDE solution required). So benchmarking our approach against the vanilla method is in effect a comparison with the scheme proposed by Han et al.

---

> > ### Comment · Reviewer_SeZL · 2024-11-25
> >
> > Thank you for your response. However, it appears that the authors have referred me to external material without making an effort to address my specific concerns or provide insights of their own. For instance, regarding the proof of Proposition 1, the authors merely stated, "Proposition 1 is a direct consequence of Girsanov's theorem, as our proof shows," without directly addressing the questions I raised about the proof.
> >
> > As a result, I will maintain my score and vote for rejection.
> >
> > PS: I believe machine learning conferences like ICLR might not be the most suitable venues for evaluating the merits of this work. I recommend that the authors consider submitting their work to top-tier journals such as IEEE Transactions on Automatic Control or SIAM Journal on Control and Optimization if gets rejected. These venues might provide more constructive and domain-specific feedback.

---

### Official Review · Reviewer_Twu5 · 2024-10-25

**Soundness:** 3
**Presentation:** 2
**Contribution:** 2
**Rating:** 5
**Confidence:** 2

**Summary:**

This paper propose a simulation-free algorithm for the solution of generic problems in stochastic optimal control (SOC), simulating trajectories using an actual control, which is more efficient and scalable than the vanilla method commonly used recently. By applied on some examples, the paper illustrates the efficacy of the new method.

**Strengths:**

1. This paper solves a critical research problem about simulation-free stochastic optimal control problem.

2. This paper is clear, with a clear description of the methods and of the state of the art. And the presentation of the technical discussion is accurate and well-organized.

3. The organization of the evaluation sections is clear, and the presented results show the advance and efficiency of the proposed method.

**Weaknesses:**

1. This algorithm is required that the base distribution used in the generative model be a point mass distribution, which restricts the applications.

2. This article gives fewer practical examples and does not allow the reader to quickly understand the application scenarios and advantages of this method.

3. The term “simulation-free” as proposed in the title and abstract is not fully explained and may be difficult to understand for readers unfamiliar with the field and terminology.

**Questions:**

1. As seen in weakness 3, can you explain what “simulation-free” means, what are the advantages in practice, and what are the applications?

2. The paper presents 6 propositions, is it possible to describe the relationship between the propositions of each part? and their role in the methodological modeling of this paper?

3. The pseudo-code for the algorithm is listed in Algorithm 1, but is it possible to create a diagram of the algorithmic framework that represents in detail the operation of the network to make it easier for the reader to understand?

4. In the comparision experiment “Quadratic Ornstein-Uhlenbeck Example (easy setup, no warm-start)” in Fig. 2 right panel, the vanilla method seems not converge. Can a convergent method be compared with the proposed method?

5. The experiment in Fig. 4 shows an example of the method with heat map in a region constant with grid. Is the method able to handle with high-dimensional problems since the grids-based example?

---

> ### Author Response · Authors · 2024-11-25
> **Response to Reviewer Twu5 (Part I)**
>
> We thank the reviewer for their constructive comments. Here is our reply and we hope this helps address the concerns of the reviewer.
>
> **Weakness 1 (base distribution):** We do not assume that the base distribution is a point mass, i.e. *Propositions 1 and 2 hold also when the initial condition for the SDE is a nonsingular measure $\mu_0$.* The point mass assumption is only used in Proposition 3, where we consider the application of our method to the construction of Föllmer processes (i.e. Schrödinger briges between a point mass an a target), and in Proposition 5, we we consider the problem of fine-tuning a difusion. In particular, the numerical examples discussed  in Sec. 3.1 and 3.2 involve Gaussian initial data $X_0 \sim N(0,\frac12 d)$.
>
> **Weakness 2 (application scenarios):** In terms of applications, our method can be applied to any scenario that fits in the formalism described in Section 2.1. Concretely, in the original submission, we illustrated the method on two types of examples: the control of Ornstein-Uhlenbeck processes, and the sampling from a target distribution with unknown normalization factor (see Sec. 3.3). In the revision, we will include another, more challenging, example (see Section 3.4), coming from statistical lattice field theory. More precisely, we show how our method can help fine-tuning a generative model to sample from the so-called $\varphi^4$ model.
>
> **Weakness 3 (simulation-free):** There are indeed several ways to understand this terminology. Here, we mean that *we need to simulate the data but we do not need to backpropagate through trajectories when computing the gradient*. In the original submission, we had written that our simulation-free approach *"avoids the extra cost of the differentiation through the solution of the SDE".* In the revision, we have added the following sentence in the introduction: *"We refer to this approach as simulation-free."* We hope this will avoid any confusion. We would also consider changing the title if you think this could help.
>
> **Question 1 (simulation-free):** See our answer to **Weakness 3** above.
>
> **Question 2 (relationship between propositions):** Thank you for giving us the opportunity to clarify the connections between our theoretical results. There are 3 groups of 2 results.
> - The first two results give the foundation for our algorithm. **Proposition 1** gives an expression for the gradient of the loss function $L(\theta)$ which involves only $\partial_\theta u^\theta$ and does not require differentiating through the trajectory $X^\theta$. **Proposition 2** reformulates this gradient in terms of an alternative loss, $L(\theta,\bar\theta)$ which makes the computation readily implementable by automatic differentiation. This result is the foundation for our algorithm; see Eqs. (11) & (12).
> - The next two propositions are about sampling from a distribution using a Föllmer process: **Proposition 3** rephrases the problem as an SOC problem, and **Proposition 4** explains how to correct the generated samples by reweighting in order to obtain unbiased estimators for expectations.
> - The last two results, **Propositions 5 and 6**, are generalizations of Propositions 3 and 4, respectively, to finte-tune a generative model.
>
> **Question 3 (Algorithm 1):** In the revision, we have clarified how we carry out all the computations in Algorithm 1.  Here's a table of our method as compared to the existing methods.
>
> |  | Avoid Back-propagation| Avoid Adjoint Method|
> | --- | ----- | ---|
> | **Our method** | Yes | Yes
> | **Vanilla Method** | No | Yes
> | **Adjoint Matching** [1] | Yes | No
> | **SOC Matching** [2] | Yes| No
> | **CMCD** [3,4] | No |Yes
>
> [1] Domingo-Enrich, C., Drozdzal, M., Karrer, B., \& Chen, R. T. (2024). Adjoint matching: Fine-tuning flow and diffusion generative models with memoryless stochastic optimal control. arXiv preprint arXiv:2409.08861.
>
> [2] Domingo-Enrich, C., Han, J., Amos, B., Bruna, J., & Chen, R. T. (2023). Stochastic optimal control matching. arXiv preprint arXiv:2312.02027.
>
> [3] Berner, J.,  Richter, L. & Ullrich, K. (2022). An optimal control perspective on diffusion-based generative modeling.  arXiv preprint arXiv:2211.01364
>
> [4] Vargas, F., Padhy, S., Blessing, D. Nüsken, N. (2023). Transport meets Variational Inference: Controlled Monte Carlo Diffusions. arXiv preprint arXiv:2307.01050

---

> ### Author Response · Authors · 2024-11-25
> **Response to Reviewer Twu5 (Part II)**
>
> **Question 4 (Fig. 2 right panel):** To confirm the convergence of the result shown in the right plot of Figure 2, we ran both algorithms for a much longer computation time and we plotted the results with confidence intervals. The new plot can be found in the [anonymous GoogleDrive](https://drive.google.com/drive/folders/1BHguocVU_jcxl_lyXg6bCYST3wg2Snpd?usp=sharing) and wil also be included in the revised version of our paper. We found out,  from the left panel of Figure 2, that both algorithms converged because the $L^2$ error on the control is very small. In this specific problem, although we initialize the neural network's parameters randomly, the initial control (before the first episode) is already quite good, with a control error of order $10^{-1}$ and this is why there is not room for much improvement in the empirical loss function plotted in the right panel of Figure 2.
>
>
> **Question 5 (Fig. 4):** The grid only appears in the plot, not in the computations. This is a ten-dimensional numerical example where the conditional distributions of $x_1,\ldots,x_9$ all depend on $x_0$. The plots in Figure 4 are made by squeezing these 9 dimensions (i.e., $x_1,\ldots,x_9$) into one single dimension and plotting the histogram of the new dimension with the dimension of $x_0$. The simulations and the neural network's inputs are in continuous space, which allows to deal efficiently with high dimensions.
>
> We thank again for the careful review and the invaulable suggestions made by the reviewer. Given these additional details we have provided about our work, we respectfully ask the reviewer to consider raising the score to better reflect our work's contributions.

---

> > ### Comment · Reviewer_Twu5 · 2024-11-29
> >
> > I appreciate the author’s detailed response. Considering my limited expertise in the field and the perspectives of the other reviewers, I regretfully adjust the contribution score to 2 (fair), the rating to 5, and confidence level to 2.
> >
> > If the novelty concerns are adequately addressed, the rating may be reconsidered.

---

### Official Review · Reviewer_bnzD · 2024-11-01

**Soundness:** 3
**Presentation:** 2
**Contribution:** 1
**Rating:** 3
**Confidence:** 4

**Summary:**

The paper considers a likelihood ratio gradient estimator for stochastic differential equations with a controlled drift. The goal of the proposed estimator is to compute the gradient of a cost function consisting of a cumulative cost part and a terminal cost. The authors claim that the "novelty" of their approach lies in the simulation-free (but the on-policy simulation of the $X^\theta$ process is still needed) nature of their algorithm, which doesn't need to simulate the sensitivity processes of $X_t^\theta$ w.r.t. $\theta$. The authors then demonstrate that the computation time and memory cost of the likelihood ratio gradient estimator is improved over the solver back-propagation method.

**Strengths:**

The notation and writing are clear.

**Weaknesses:**

Unfortunately, I do not see any novelty in the method proposed in this paper... The likelihood ratio gradient estimator for SDEs is well-known (Yang & Kushner, 1991). I am uncomfortable with the authors' claim that this approach is their contribution.

I am open to further discussion, but in order for me to raise the score, please explain the novelty of your method or the contribution beyond implementing this estimator. I am okay with experimental/empirical validation of the effectiveness of the LR gradient estimator in complex machine learning applications, but the current version of the paper clearly isn't able to address this.

Moreover, the LR-based methods, due to the nature of laws of SDEs, are only applicable to settings where the drift is controlled; i.e. the volatility cannot depend on $\theta$. Furthermore, I believe that it is usually the case that the LR method has a significantly larger variance in settings where the volatility is small (because you are forcing the diffusion to go to unlikely locations), and one could imagine that optimal control prefers parts of the state space where the volatility is indeed small. So, this seems to suggest that the variance performance, which is not compared in this paper, will be worse compared to back-prop or pathwise sensitivity-based methods like Li et al. 2020, Massaroli et al. 2021, and Wang et al. 2024.

> Massaroli, Stefano, et al. "Learning stochastic optimal policies via gradient descent." IEEE Control Systems Letters 6 (2021): 1094-1099.


>Li, Xuechen, et al. "Scalable gradients for stochastic differential equations." International Conference on Artificial Intelligence and Statistics. PMLR, 2020.



>Wang, Shengbo, Jose Blanchet, and Peter Glynn. "An Efficient High-dimensional Gradient Estimator for Stochastic Differential Equations." arXiv preprint arXiv:2407.10065 (2024).

**Questions:**

There seems to be a typo in equation (10).

What happened to the right one in Figure 2? Are you using different sample sizes to compute the empirical losses? You should plot a confidence interval instead of these lines: the variability in this plot does not indicate a worse variance performance of the estimator, it just suggests that you are using fewer sample paths to evaluate the empirical cost. Am I right?

How well does the estimator perform in terms of variance?

---

> ### Author Response · Authors · 2024-11-25
>
> We thank the reviewer for their constructive comments.
>
> **Weakness 1 (novelty):** Just before Proposition 1, we clearly stated that *"A similar formula first appeared in Yang & Kushner (1991) and it is also given in Ribera Borrell et al. (2024)"* so we do not claim this is our contribution. We will rephrase how we introduce Propostion 1 to clarify this.
>
> The main novelty of our work is the implementation of an eficient algorithm using this formula for the solution of SOC problems via deep learning, with application to recent problems such as the construction of Föllmer processes and the fine-tuning of generative models. These problems have recently generated a lot of interest in the machine learning community, and for this reason we believe that our results will be of interest to, and generate more work in, this community.
>
>
> **Weakness 2 (applicability):** We agree that our method is not directly applicable to settings where the volatility is also controlled. However many SOC problems, including those arising in machine learning, have only a controlled drift (e.g. fine-tuning diffusion models [1], Schrodinger Bridge problems [2]). Furthermore, we believe there are ways to extend our method to controlled volatility, for example by using Malliavin calculus, following the lines of (Gobet & Munos, 2005). We mention this point in the revision after Proposition 1.
>
>
> **Question 1 (typo):** There was indeed a typo in the first term, which should be multiplied by $1/2$. We have corrected it in the revision. Our code implements the correct formula, which is updated in the revised draft.
>
> **Question 2 (Figure 2):** Actually we ran both algorithms with the same mini-batch size. To confirm that there is no mistake in the result shown in the right plot of Figure 2, we ran both algorithms for a much longer computation time and we plotted the results with confidence intervals. The new plot can be found in the [anonymous GoogleDrive](https://drive.google.com/drive/folders/1BHguocVU_jcxl_lyXg6bCYST3wg2Snpd?usp=sharing).
>
> **Question 3 (variance of estimator):** The norm of our gradient estimator might have a larger variance than the one of the vanilla method, but it is actually not so easy to tell *a priori* wherher that is the case or not. In addition, and more to the point, what controls the actual performance of the algorithm is the variance in the loss function or the $L^2$ error.  From the point of view of these metrics, our experiments (including the new ones available in the [anonymous GoogleDrive](https://drive.google.com/drive/folders/1BHguocVU_jcxl_lyXg6bCYST3wg2Snpd?usp=sharing)) indicate that our method performs very well, and better than the vanilla method. If you have a specific concern in mind, please feel free to give us more concrete way to convince you.
>
> [1] Domingo-Enrich, C., Drozdzal, M., Karrer, B., \& Chen, R. T. (2024). Adjoint matching: Fine-tuning flow and diffusion generative models with memoryless stochastic optimal control. arXiv preprint arXiv:2409.08861.
>
> [2] Liu, G. H., Lipman, Y., Nickel, M., Karrer, B., Theodorou, E. A., \& Chen, R. T. (2023). Generalized Schrödinger Bridge Matching. arXiv preprint arXiv:2310.02233.

---

> > ### Comment · Reviewer_bnzD · 2024-11-28
> > **Response**
> >
> > I thank the author for the clarification. As I have explained, I do not think identifying an existing method and applying it in some small-scale experiments is a enough contribution. If the goal of the paper is to demonstrate that the likelihood ratio gradient estimator in Yang & Kushner (1991) is effective in machine learning contexts, then the author should use this method in an environment where there are real data or real benchmarks.
> > For example, one can attempt to fit financial data using neural SDEs, or solve PDE constraint optimization using the gradient method and compare it to the existing neural network-based PDE solver and finite element approach.

---

> > > ### Author Response · Authors · 2024-11-28
> > >
> > > We believe that the reviewer's assessment is harsh and puts the bar too high.
> > >
> > > Kushner and Yang's formula for the gradient of the SOC objective is not well known, let alone used, by the ML community. This is a pity in view of the growing interest in SOC problems in this community, especially in the context of generative AI. *It would also not be the first time that results previously known would be given a second wind motivated by ML applications*: examples of such technology transfer have been very successful in the past, e.g. in the context of diffusion models in general and Schroedinger bridges in particular (an instance of which we study in our paper).
> > >
> > > We would also like to point out that our numerical examples are standard benchmarks that have been used in recents papers on SOC, e.g. [1,2,3,4], and that our results show that *our method outperforms existing ones in all instances*. Our results, including the new ones that we have included in the revised version of our paper, also show that the results from our approach have *less variance*  than those obtained with the vanilla method: this addresses a concern raised by the reviewers, and indicates not only that our method is viable but also that the negative a priori against using Kushner and Yang's gradient formula because of its variance is probably ill-founded (another point well-worth making to the ML community).
> > >
> > > Finally, let us say that we intend to perform experiments on larger scale examples, but we will need to do this later on, in collaboration with other groups. The reason is simple: as the reviewer is well-aware of, such experiments are typically very demanding and require a level of computing power we do not have access to. We believe we should not be discriminated against because of this, especially since we believe that our results already indicate that the approach has mileage.
> > >
> > > [1] Nikolas Nüsken and Lorenz Richter. Partial differential equations and applications, 2:1–48 (2021)
> > >
> > > [2] Qinsheng Zhang, Yongxin Chen, arXiv:2111.15141 (2021)
> > >
> > > [3] Carles Domingo-Enrich, Jiequn Han, Brandon Amos, Joan Bruna, Ricky TQ Chen, arXiv:2312.02027 (2023)
> > >
> > > [4] Carles Domingo-Enrich, Michal Drozdzal, Brian Karrer, Ricky TQ Chen, arXiv:2409.08861 (2024)

---

### Official Review · Reviewer_2UrK · 2024-11-09

**Soundness:** 2
**Presentation:** 3
**Contribution:** 2
**Rating:** 6
**Confidence:** 4

**Summary:**

The paper, titled "A Simulation-Free Deep Learning Approach to Stochastic Optimal Control," proposes a novel method for solving high-dimensional Stochastic Optimal Control (SOC) problems using a deep learning-based approach. The prposed approach leverages the Girsanov theorem to compute gradients on-policy without needing adjoint-based differentiations. The method is shown to handle complex SOC scenarios more efficiently in terms of memory and computational resources.

The method is shown to handle complex SOC scenarios more efficiently in terms of memory and computational resources. However, its reliance on simulation-free gradient estimation may introduce variance issues when applied to challenging SOC landscapes, where maintaining accuracy across high-dimensional distributions without simulations may be difficult.

**Strengths:**

1. The use of the Girsanov theorem for gradient estimation is theoretically sound, particularly in reducing the computational burden traditionally associated with Neural SDEs. This efficiency makes the approach suitable for high-dimensional SOC problems and applications that involve large neural networks.

2.  This formulation allows for gradient calculations without differentiating through the SOC trajectories, which is a considerable advancement over existing SOC methods.

3. Experimental results show clear computational benefits in terms of memory usage and runtime, as well as improved accuracy over baseline methods such as Neural SDEs and the Path Integral Sampler.

**Weaknesses:**

1. The reliance on the Girsanov theorem and assumptions related to Wiener processes and Gaussian distributions may limit the approach's applicability to more general stochastic processes or non-Gaussian noise environments.

2. While the method shows improvements in certain metrics, the paper does not discuss potential instability or convergence challenges in high-dimensional settings or on extended time horizons.

3. While the proposed approach avoids back-propagation through stochastic differential equations, it introduces complexity in high-dimensional SOC problems. This limitation could hinder scalability in large-scale applications.

**Questions:**

1. The simulation-free gradient estimation may introduce variance issues. How does the method handle situations where high variance arises in the gradient estimates?

2. The approach relies on the reference process being close to the target control. Could the authors elaborate on how to select or adapt this reference process dynamically?

---

> ### Author Response · Authors · 2024-11-25
>
> We thank the reviewer for their constructive comments. Here is our response to the concerns of the reviewer.
>
> **Weakness 1 (Girsanov theorem):** While we agree that our approach relies on some assumptions about the form of the control problem, we would like to stress that there are many interesting problems that fall in the framework covered by our assumptions.  In our paper we focus on applications involving the construction of Schrödinger bridges and the fine-tuning of diffusion models since they are extremely important in recent applications of generative AI. We would also like to stress that Gaussian noise is ubiquitous in a wide range of applications, including again diffusion models for instance. It might possible to extend our method to more general forms of noises and controlled dynamics but we leave this for future work.
>
> **Weakness 2 (high dimension and time horizon):** While the computational time increases with dimensionality and time horizon, as with any numerical methods, we have not observed particular challenges with our method due to these aspects. To demonstrate this, we have added one more experiment of the Quadratic OU type, with $d=50$ and $T=4$. Please check the results in this [anonymous GoogleDrive](https://drive.google.com/drive/folders/1BHguocVU_jcxl_lyXg6bCYST3wg2Snpd?usp=sharing).
>
> **Weakness 3 (complexity in high dimension):** We are not sure what type of *"complexity"* you are referring to. Computing the gradient of the SOC objective via our Proposition 2 is much less complex than doing so via backpropagating through the whole trajectory (even in high dimension) because it can be done timestep-per-timestep. We have clarified this in the new version of Algorithm 1, which explains in detail how the computations are done. Please us know if you would like further clarification.
>
> **Question 1 (high variance):** We have not observed such issues in our experiments. The norm of the gradient might have a large variance but this does not necessarily impact the performance of the algorithm. What really matters is the variance in the loss function or the $L^2$ error. From the point of view of these metrics, our experiments show that our method performs very well, with *lower* variance than the vanilla method. If you have a specific concern in mind, please feel free to give us more concrete way to convince you.
>
> **Question 2 (target control):** We are not sure what you mean by *"reference process"*. If you refer to the *"reference control"* denoted by $v$ in Lemma 1, note that this control is set to the *current* control, $v = u^{\bar\theta}$, in Propositions 1 and 2: that is,  in our algorithm, $v = u^{\bar\theta}$ can be viewed simply a version of the neural network $u^\theta$ with respect to which we do not compute the gradients. In practice, we have a single neural network but we use stopgrad on the terms involving $v=u^{\bar\theta}$ to detach them from the compuational graph and prevent their gradients from being computed. This $v=u^{\bar\theta}$ is also used to generate trajectories. There was a typo in the code and we corrected it in the revised version.
>
> We also stress that $v = u^{\bar\theta}$  does not need to be close to the *target* (optimal) control, at least not initially: it simply converges towards this target during training.
>
> Given these additional explanations about our work and the experimental evaluations we have added, we respectfully ask the reviewer to consider raising the score to better reflect our work's contributions.

---

### Author Response · Authors · 2024-11-24
**General Response to Reviewers**

We thank the reviewers for their careful reading of our paper and for ackowledging that our method brings "considerable advancement over existing SOC methods" and "the presented results show the advance and efficiency of the proposed method". Their constructive comments helped us to improve our paper. We reply to each reviewer separately below but we first address some common concerns in this general response:

**Scalability**: Our method remains competitive even in high dimension. To illustrate this point, we have included two new examples in higher dimension ($d=50$ and $d=16 \times 16 = 256$).

**Novelty**: We have clearly stated that the formula appearing in Proposition 1 was originally derived by Yang & Kushner (1991). The main novelty of our work is to leverage this formula for the efficient solution of SOC problems with deep learning. We also illustrate the usefulness of our approach in applications such as the construction of Föllmer proceses and the fine-tuning of generative models, which have received considerable attention lately in the machine learning community.

**Variance of the estimator**: While it may be true that the norm of our estimator has a larger variance than in the vanilla case, this is not what matters in practice. The important quantities to measure performance are the loss and, when available, the $L^2$ error on the control. For these metrics, our method performs better (faster convergence in terms of computational time, and less variance in most cases), as illustrated by our numerical examples.

**Range of Applications**: Our method does not work for SOC problems in which volatility is controlled or parameterized. This still leaves room for many applications of interest in which the volatility is fixed, including  the solution to Schrödinger Bridge problemss or the fine-tuning of pre-trained diffusion models. We have included more numerical examples of this sort in the revision. In addition, we believe there are ways to extend our method to controlled volatility, for example by using Malliavin calculus, following the lines of (Gobet & Munos, 2005). We will also mention this point after Proposition 1 in the revision.

---

### Meta-Review · Area_Chair_xoiT · 2024-12-21

**Metareview:**

In this work, the authors propose a simulation-free algorithm for solving generic stochastic optimal control problems using applications of Girsanov's theorem, and gradient results from Yang & Kushner (1991). Benchmarks appear to improve upon other adjoint method approaches. Unfortunately, the paper does not appear to provide a sufficiently significant contribution. I understand the value in bringing light to results from other literatures, but this should not provide low-hanging fruit for submission to top-tier ML outlets. The method here is not new, and the benchmarks are not only synthetic, but also similar to those considered previously so no novel capacity is demonstrated either. It does not matter if the method performs better than those within the popular ML outlets; this is not new material.

I would recommend that the authors perform significantly more advanced real-world experiments to demonstrate that this method provides improved capacity and value at the frontier of the field. The authors mentioned in the discussion phase that this requires significant compute and collaborations with other researchers. I believe that is the bar. Otherwise, further theoretical contributions are necessary to compensate for this effort.

**Additional Comments On Reviewer Discussion:**

All reviewers provided mixed to negative scores. Reviewer bnzD raised the concern of novelty, and the authors responded with some justification as to why they felt the contribution was still significant in light of improved benchmarks. The reviewer did not agree with this sentiment. Reviewer Twu5 raised concerns regarding the presentation of the work (particularly the terminology 'simulation-free'), and the small number of examples. Reviewer Twu5 reduced their score, agreeing with the other concerns of novelty. Reviewer SeZL raised a few "critical errors" in the paper, which the authors disagreed with and felt the reviewer did not understand the work. The reviewer was dissatisfied with the response and the discussion was left at an impasse.

---

### Decision · Program_Chairs · 2025-01-22

Reject